# Economic valuation of natural promenades in Iran using zonal travel costs method (Case study area: Gahar Lake in Lorestan Province in western Iran)

Ebrahim Kheyri[1], Maryam Morovati [2,3]*, Akram Neshat[4], Gholamreza Siahati[2]

**1** MSc Assessment and Land Use Planning, Faculty of Agriculture and Natural Resources, Ardakan University, Ardakan, Iran, **2** Department of Environmental Sciences & Engineering, Faculty of Agriculture & Natural Resources, Ardakan University, Ardakan, Iran, **3** Medicinal and Industrial Plants Research Institute, Ardakan University, Ardakan, Iran, **4** Department of Agricultural Economics, Faculty of Agriculture & Natural Resources, Ardakan University, Ardakan, Iran

* mymorovati@ardakan.ac.ir

## Abstract

Gahar Lake is located within Oshtorankooh Protected Area (east of Lorestan Province in Iran), which has extensive potentials for the development of the tourism industry. The aim of the present research was to determine the economic value of the Gohar Lake resort using the zonal travel cost method. Therefore, at first, 380 questionnaires were distributed among the tourists by the simple random sampling method based on appropriate spatiotemporal distribution during the visiting seasons. The questionnaire items were categorized as economic, social, and miscellaneous parts. The calculation results revealed a value of USD 84.538 per visitor and a value of USD 1,986,657.163 per year, indicating the high value and importance of the region. The analysis showed that socio-economic variables have a significant role in the use or non-use of the resort. The obtained R2 coefficient was 0.82, indicating that around 82% of the changes in the number of visitors can be justified by the variables introduced in the model. The results also revealed the need to pay more attention to this region and formulate a tourism development plan.

## Introduction

The environment and natural ecosystems provide goods and services which, regardless of whether or not they are used, ultimately lead to the well-being of communities [1, 2]. According to the definition of the World Tourism Organization, tourism is a set of activities persons do, while traveling to or staying in a place other than their own living places for the purposes of entertainment, recreation, rest, sports and so on, which do not last for more than one year [3, 4]. Valuing tourist areas and environmental functions is an important step in modifying economic decisions, which often assume environmental resources as free goods and services. Economical valuation is a tool to distinguish between efficient and inefficient uses of goods and to realize the true value of natural resources in the decision-making process of

**Funding:** The authors received no specific funding for this work.

**Competing interests:** The authors have declared that no competing interests exist.

environmental management [5, 6]. The total economic value of ecosystems can be divided into use and non-use values.

The use values generally require real consumption, and in contrast, non-use values occur without any direct interference with their natural environment [7]. Accordingly, the valuation of non-market functions and services of natural resorts would be of utmost importance for many reasons, including "recognition recognizing and understanding the environmental and ecological benefits", "making the decision-makers and planners familiar with the environmental issues", "making links between economical economic policies and natural incomes", "assessing the role and importance of environmental resources in supporting human welfare and sustainable development", "adjusting and modifying national metrics such as Gross domestic product (GDP)", and "preventing the degradation and overexploitation of natural resources" [8–10]. The recreational value, which is among the use values of a natural system include the use of forests or natural resources for recreational purposes, leisure and entertainment, hiking, mountaineering, fishing, and aesthetics. Economic valuation of the environment based on its application to tourism prevents many of the developmental options which threaten the environment [11]. Nowadays, many of the well-known benefits of lakes are in the form of non-market goods such as water quality and wildlife protection, which, although important to the community, receive less attention.

Ignoring the market value of these services means they are often underestimated in terms of their degradation or their private economic benefits. However, the studies to quantify the value of lakes are highly controversial given their diversity, socioeconomic contexts, and complex biological functions of water. A policy is needed to bring the market and non-market values of lakes into interaction in decision-making processes [12–15].

Various studies have been carried out using the zonal travel cost method to estimate the recreational value of different regions. As such, Ezzy and Scarbrough [16] studied the recreational value of bluefin tuna fishing in Portland, Australia, using the zonal travel cost method and reported the recreational value of the region between USD 132 and USD 133 per person per visit and USD 499,533 and USD 1,325,124 per year [16]. Park et al. [17] also examined the feasibility of renewable energy as a substitute for nuclear energy. The results of this study indicated that an average household is willing to pay the extra cost (about USD 85) per month. The value of renewable energy in Korea is about KRW (Korean Republic won) 19.3 trillion (approximately USD 16.1) [17]. Similarly, Rewitzer et al. [18] in a research on the economic value of cultural ecosystem services of Alp landscape in Switzerland, used the travel cost method and reached the conclusion that willingness-to-pay was nearly EUR 80 for the additional 60 hectares of dry grassland annually [18].

Since Gahar Lake enjoys features such as diverse natural landscapes, high diversity of wildlife species, the existence of rare and endangered animal species, diverse flora, and being relatively wilderness, so it was selected for a case study in this research. The aim of this study was to determine the economic value of the lake resort area using the zonal travel cost method. The study of the economic value of natural resorts is among the topics that have been in the focus of scholars in recent years. Since natural areas, each has its own set of site-specific characteristics; the study of each area would somehow be innovative. This is the first time that the valuation study for Gahar Lake in Lorestan Province is carried out at this sample volume and by this method.

## Materials and methods

### Study area

Gahar Lake with geographical coordinates of 49°16′40″ to 491748″ eastern longitude and 33°18′12″ to 33°18′44″ northern latitude is located in the middle of a narrow valley between two the ranges of Oshtrankooh Mountains in the Oshtrankooh Protected Area (Fig 1).

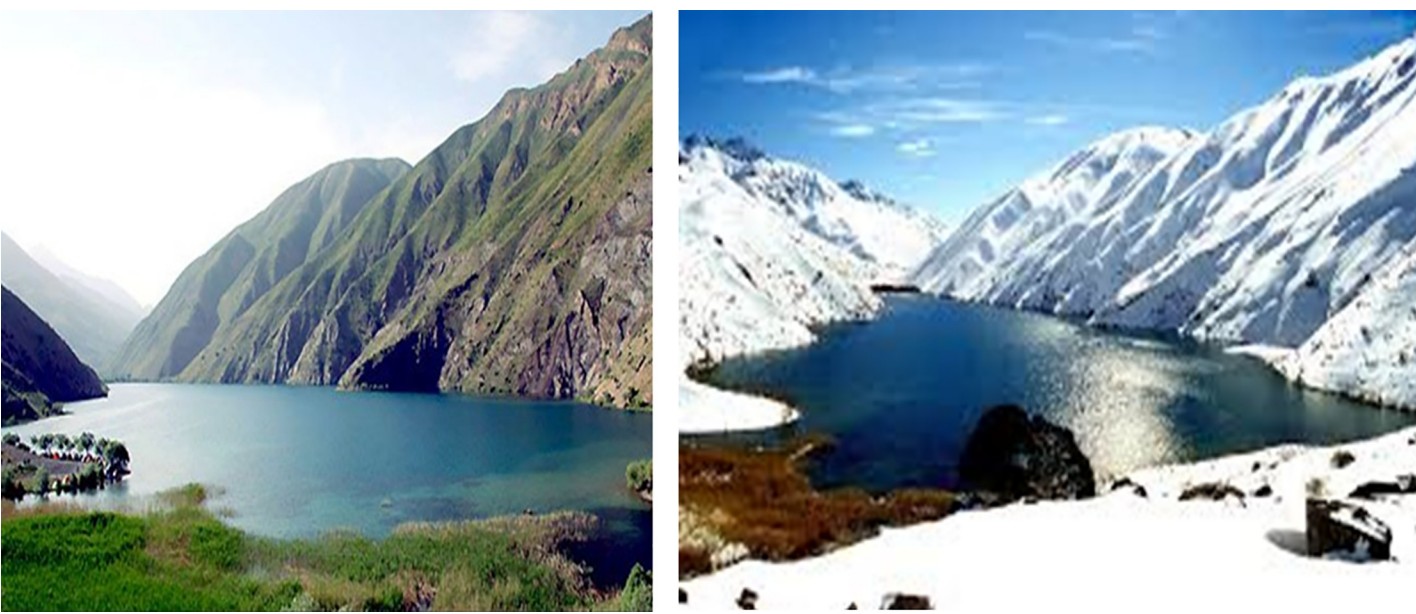

**Fig 1. Gahar Lake in different seasons.**

This lake has been formed in the mountainous bed of Gahar River, at an altitude of 2350 m above sea level and along a fault valley. Gahar River, which hosts the lake, is one of the main tributaries of the Sezar River, eventually joining the Dez River. The lake has a distance of 121 km from Khorram Abad, the province capital city, 35 km from Doroud County, and 18 km from the closest village of Darb Astaneh (Fig 2). Gahar River flows from the southeast to the northwest in its upper bed. After the formation of Gahar Lake in the Gahar River valley. It

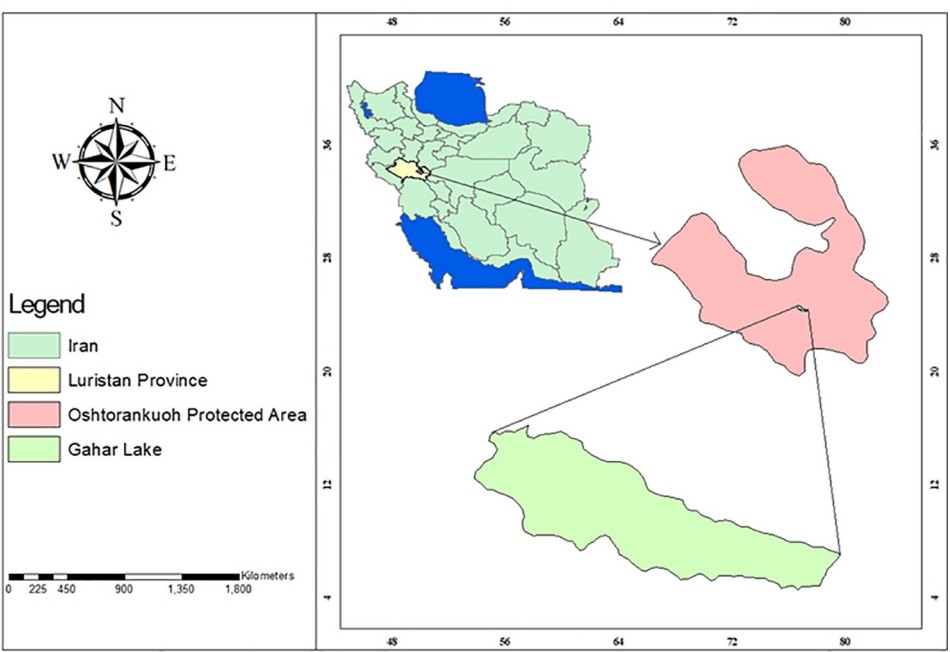

**Fig 2. Location of Gahar Lake in Oshtrankooh Protected Area in Lorestan Province and Iran.**

changes direction in the north of Takht-e Kala Mountain, flows to the east along "Tang-e Helilan" and after reaching Haftkuh, it turns to the northwest again. In parallel to the same valley after crossing in front of the villages of "Emarat" and "Baraftab", it flows into the Sezar River. Gahar Lake is composed of two sections, the great Gahar or Kalleh Gahar and the small Gahar or Koreh Gahar. In both parts of the lake along the course of Gahar River, due to landslide movements and instability of the slopes, stone and soil masses fall into the riverbed. According to the studies, the area of Gahar Lake is around 85 km and the area of its basin is around 52 km$^2$. The maximum depth of the lake is nearly 28 m and the minimum is 3 m. More than half of the lake is more than 20 m deep [19].

Oshtrankooh Protected Area with an area of 104355 ha is located in the south and southeast of Doroud City in the west of Azna City. Access to this region is by Doroud-Azna Road, which passes through the city. The minimum and maximum altitudes are approximately 1400 m in Chamchid and 4090 m in Sanboran Mount (Oshtrankooh). The hydrographic network of the study area includes numerous perennial and seasonal rivers, watercourses, and canals, which mostly originate from the heights of Oshtrankooh. Gahar River is the most important river in the Oshtrankooh Protected Area [20].

### Research methodology

The method used to determine the recreational value of the study area was the zonal travel cost. In this research, which was conducted in 2018, 380 questionnaires (determined using the Cochran's formula) were distributed among the tourists using the simple random sampling method based on the appropriate spatiotemporal distributions during the visiting seasons. The questionnaire items were categorized into 3 groups of economic, social, and miscellaneous. Statistical calculations and analysis of the findings were conducted using Eviews software version 8 and EXCEL. The following steps were taken to perform the zonal travel cost method:

Step 1: Preparing the map of the study area and positioning the visiting sites

Step 2: Drawing concentric circles with different radii and fixed distances in order to assess the distance parameter.

Given that, the travel cost method is based on estimating the relationship between the number of visitors to a resort, the distance of visitors' living place, and the average cost of access, so the relationship between the visitors' residence and their distance from the resort should be determined. Therefore, using the map of Iran and considering the resort as the center, 8 circles were drawn at fixed distances of 150 km so as to well-cover the entire country.

Step 3: Calculating the resident population in each zone. The resident population in each of the zones was calculated according to the country's latest population and housing census in 2016.

Step 4: Estimating the "number of visits to the site from each zone within a specified time frame", "average cost of access from each zone to the resort", and" average distance of resort from each of the zones enclosed in the above-mentioned circles". The age groups of the visitors were considered in five groups, the education level in 6 groups, and the monthly income in 6 groups (Tables 1 and 2).

Step 5: Modeling and calculating the relationship between the number of people visiting the resort and the distance parameter as well as the access cost using Clawson travel cost method based on the following equation:

$$V_{ij}/N_i = F(TC_{ij}, S_{ij}, A_{jk}) \tag{1}$$

Where; $Vij$ is the number of visitors from the Zone i in the resort j, $N_i$ is the total population of the Zone i, $TC_{ij}$ is the cost of access to the Resort j from the Zone i, $S_{ij}$ is the socioeconomic

**Table 1. Education level of the visitors.**

| Education level | Number of visitors | % |
|---|---|---|
| School dropouts | 11 | 2.9 |
| Diploma | 80 | 21.1 |
| Associate | 121 | 31.8 |
| Bachelor | 107 | 28.2 |
| Master | 50 | 13.2 |
| Ph.D | 11 | 2.9 |
| | 380 | 100 |

characteristics of the people who live in the Zone i and use the Resort j, and $A_{jk}$ is the attractive and aesthetic features of the resort j compared to the other recreational sites (k).

Step 6: plotting the demand curve using the relations obtained for the resort.

Step 7: Calculating the area under the demand curve as the economic value of the resort:

The least-squares method was used to estimate the demand function and calculate the area under the curve in this function. In this model, the dependent variable was the number of visitors per 1000 zonal population and the independent variables were age, education level, and average monthly income.

Step 8: Case study of some characteristics of the visitors, such as education level, age, gender, income level, etc.

In order to observe the ethical protocols, before distributing the questionnaires, the research objectives were explained orally to the participants. They were rest-assured that participating in the survey was completely voluntary and that they could easily resign without any reprimand if they did not wish to continue cooperating after studying the questions or at any stage. The questionnaires were anonymous and participants were assured that the information would keep strictly confidential. The work was carried out after obtaining the necessary permission from the Department of Environment under the supervision of the park manager. This poll was only for adults and minors did not include to the study.

## Results

### Relationship between zones and number of visitors

Table 3 shows the number of visitors coming to the region from each zone. As the table suggests, Zones 1, 3, and 2 respectively accounted for the highest number of visitors. With a dramatic difference, Zones 4 and 5 are placed next, from where 34 and 11 visitors visited the site. Except for Zones 7 and 8 with no visitors, Zone 6 has the lowest share (1.8%) of the total visitors.

**Table 2. Average monthly income of the visitors.**

| Average income | Number of visitors | % |
|---|---|---|
| 98.034 | 19 | 5 |
| 142.551 | 38 | 10 |
| 190.068 | 110 | 28 |
| 285.103 | 76 | 20 |
| 356.379 | 78 | 20.5 |
| 475.172 | 59 | 15.5 |
| Total | 380 | 100 |

**Table 3. Number of visitors from each zone.**

| Zone | Number of visitors | % | Per 1000 people |
|------|--------------------|------|-----------------|
| 1 | 138 | 36.3 | 0.0138 |
| 2 | 80 | 21 | 0.008 |
| 3 | 110 | 28.9 | 0.011 |
| 4 | 34 | 8.9 | 0.0034 |
| 5 | 11 | 2.8 | 0.0011 |
| 6 | 7 | 1.8 | 0.0007 |
| 7 | 0 | 0 | 0 |
| 8 | 0 | 0 | 0 |

There is an adverse relationship between the distance traveled and the number of trips, so the more the distance, the less the number of visitors. There is only one exception in Zone 3, which, contrary to the expectations, has more visitors than Zone 2. This may attribute to the larger population of this zone.

## Average travel cost of visitors

Fig 3 depicts the relationship between the zones and average travel cost of the visitors. According to the figure, as the traveled distance increases, the average cost of the travel also increases, and those who visit the site the most, actually pay less for the trip.

## Data analysis by Eviews software

**Linear regression.** In the first step, the variables with a negative effect on the function were removed and after eliminating heterogeneity, autocorrelation, and collinearity, the coefficients of the remaining variables were calculated as in Table 4. The socioeconomic variables of the visitors show that these variables have a significant role in influencing the use or non-use

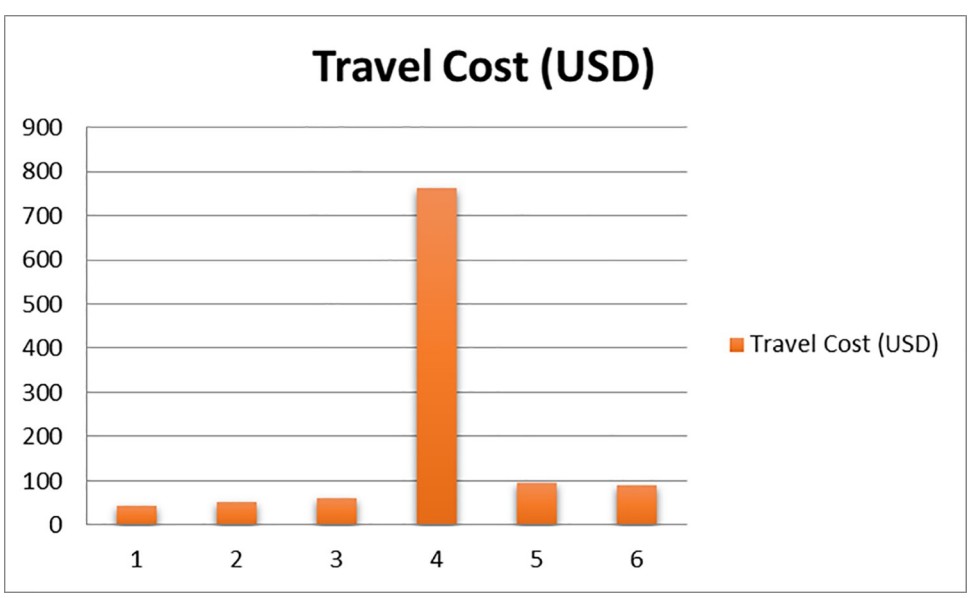

**Fig 3. Relationship between the zones and average travel cost.**

**Table 4. Coefficients of the variables.**

| Variable | Coefficient | T-statistic value | Significant level |
|---|---|---|---|
| Intercept | -9.88 | -0.89 | **0.3689** |
| Cost | -1.41 | -1.89 | **0.0591** |
| Space | - 0.00075 | 23.78 | **0.0000** |
| Average monthly income | 2.62 | 3.00 | **0.0028** |
| Education level | 2.49 | 2.02 | **0.0431** |
| Age | -3.12 | -1.60 | 0.1088 |
| **Prediction accuracy (%)** | **0.82** | **F-statistic values** | **351.35** |

of the resort. According to the estimated coefficients, a positive and significant relationship was found between the average monthly income and the use of the resort.

The coefficient of $R^2$ was obtained equal to 0.82, revealing that about 82% of the changes in the number of visitors can be justified by the variables defined in the model. The Durbin-Watson statistic, which indicates the existence or nonexistence of autocorrelation, is within an acceptable range and the obtained value shows that there is no correlation between the components of the model in question.

**Economic value of Gahar Lake.**   The recreational value of Gahar Lake was calculated based on the demand function. The regression equation of this function was calculated as follows:

$$VR = (-9.88 - 1.41TC - 0.000751S + 2.62I - 3.12A + 2.49E) \tag{2}$$

Where; VR = the number of visitors, TC = total cost, S = distance, I = = monthly income, A = age of the visitors, and E = education level. By replacing the mean value of each variable and taking an integral from the equation, which the same as the area under the demand curve, the value of the resort was determined as USD 84.538 per visitor and USD 1,986,657.163 per year.

## Discussion

Gahar Lake is one of the most beautiful natural and tourist landscapes of Lorestan Province and Iran, which is unique in its kind. This is why most tourists prefer the natural landscape of this resort to its other features. The visitors mainly intend to visit the natural attractions and landscapes of the Oshtrankooh Protected Area, especially Gohar Lake. Using the zonal travel cost method, the recreational value of Gahar Lake was calculated as USD 84.538 per visitor and USD 1,986,657.163 per year reflecting the significance and high value of this resort.

Mafigholami et al. [21] used the zonal travel cost method to value the recreational value of Kouhrang Waterfall in Iran. Based on the statistical analyses on the distribution of the visitors' origin and travel costs, the recreational value of the waterfall was estimated at USD 3,715.314 [21]. Moreover, Vakili Ghasrian et al. [22], using the choice experiment technique, determined the economic value of natural functions of Zaribar Lake and concluded that the highest final willingness to pay of USD 19.539 was related to the establishment of recreational-tourism facilities at the absolutely amended level [22].

The study of the variables such as education level, income, age and occupation as socio-economic variables of the visitors showed that they have a significant role in influencing the use or non-use of the resort. This is in line with the finding by Ofori and Rouleau in 2020 [23]. They introduced 5 parameters of income, literacy, elevation, and residence years as significant factors in willingness to pay of the tourists for the coastal ecosystems of West Africa. Tian et al. [24] claimed that those with higher socioeconomic status have a relatively higher willingness

to pay for the management of urban green spaces in central China. However, Enriquez-Ace-vedo et al. [25], in a study on willingness to pay for costal ecosystem services in three beaches in Colombian, came to a contradictory conclusion and argued that willingness to pay in the coast did not depend on the economic variables such as income or employment, while the perception variables had a decisive effect. It can be concluded that in general, economic variables can be of high importance and sometimes decisive in the willingness to pay. However, knowing and understanding the importance of ecosystem services can be just as important. People who are aware of the importance of the environment and with good financial affordability are likely to have more financial and non-financial contributions to the preservation of the environment. According to the results, 72% of the visitors were male and the rest were female. The education level of most visitors was associate's and Bachelor's degrees. In addition, the average monthly income of most of the tourists varied from USD 190.068 to USD 376.359, so the higher the income, the more visits to the resort.

The majority of the tourists were in the age group of 30–40 years, which means the visitors were mostly young. This can be due to their greater ability in passing the access routes or the incentives such as spending their leisure time in the unique natural landscape of the resort. According to the questionnaires, 54.2% of the tourists to Gahar Lake were self-employed or clerks. Most of the tourists become acquainted with this area through friends and family, which indicates poor publicity of the resort in the media. Most visitors come to this area in "group trips" and then, in the next rank, in "family trips".

The high number of group and family visitors can be due to the reasons such as the unique beauty of the landscape and its high security. Security is one of the most important and main criterion for tourists all over the world. No matter how unique and wilderness the tourist areas are, the necessary condition for attracting tourists is first and foremost the existence of security in the tourist destinations.

There was found an inverse relationship between the distance traveled and the number of trips, so the longer the distance, the lower the number of visitors. The only exception was in Zone 3, wherein the number of visitors was higher than that in Zone 2, which may be due to the more population of this zone. The travel costs also increased with the distance passed. The negative sign of the travel cost demonstrates its negative relationship with the number of visitors, i.e. the increase in travel costs leads to a decrease in the number of visitors. If visitors cannot afford to travel to this area, they will choose amusement parks outside or inside the city, and then, the forest parks outside the city. The maximum entrance cost that tourists were willing to pay for the recreational places near their place of residence was between USD 0.237 and USD 0.475; however, they were willing to pay even up to between USD 475 to USD 1,187as the entrance fee of Gahar Lake because of its superior natural landscape and other special features. The majority of them were willing to pay between USD 2.375 and USD 4.751 annually to protect the area. This willingness to pay for the conservation purposes has also been reported in other studies. As such, Xu et al. [26] found that 55.90% of the respondents have willingness to pay for conservation of green spaces in Beijing, China. Tavárez and Elbakidze in 2019 also reported that the residents in Puerto Rico have a willingness to pay up to USD 39 for the conservation of frosts. Lankia et al. [27] published the same results from Finland where 60% of the participants declared their willingness to contribute in the conservation of recreational sites through labor or direct payments. All of the above shows that people's awareness of the value of resorts is increasing and they are willing to spend to preserve these valuable resources. In other words, as information about ecosystem services and the value of ecosystems increases, more people will be willing to participate in the conservation of natural resources. Cavasos and Bhat in 2019 [28] showed that the respondent's willingness to pay strongly correlates with their perception of the anthropogenic risks of loss of coral reefs.

It should be mentioned that the amount that tourists were willing to pay for the recreational sites depended on the type and value of the region and the tourists' tastes. The services and facilities of the tourist sites had also a great impact on this amount. Unfortunately, the resort is rather poor in amenities and hygiene; however, no considerable measurement has been taken by the authorities to address these shortcomings.

## Conclusion

This study addressed the monetary value of a tourist destination in a protected area in western Iran in terms of recreational use. According to the results, the recreational value of the study area was estimated at USD 84.538 per visitor and USD 1,986,657.163 per year. It was found that the tourists have a willingness to pay up to USD 1,187 as the entrance fee. They were also willing to pay between USD 2.375 and USD 4.751 annually to protect the area. Economic and social issues were assessed to be a determining factor in this willingness to pay. Most of the tourists were young. This could be due to the poor health and welfare amenities of the resort, which makes it difficult for the elderly to use.

Conducting this study would be an important step in protecting the protected areas and natural resources and moving towards sustainable development. The high recreational value of Gahar Lake and its importance can provide justification for decision-makers, managers, and officials to preserve the quality of the resort and to plan for the optimal use of the area by improving the welfare, health, and leisure facilities in the region.

## Supporting information

**S1 Data.**
(XLSX)

**S2 Data.**
(PDF)

**S1 File.**
(RAR)

## Author Contributions

**Writing – original draft:** Ebrahim Kheyri, Maryam Morovati, Akram Neshat.

**Writing – review & editing:** Ebrahim Kheyri, Maryam Morovati, Akram Neshat, Gholamreza Siahati.

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
