## [Decision Letter · Decision Letter 0]

18 Jun 2020

PONE-D-20-11207

Economic Valuation of Natural Promenades in Iran Using Zonal Travel Costs Method (Case study area: Gahar Lake in Lorestan Province)

PLOS ONE

Dear Dr. Morovati,

Thank you for submitting your manuscript to PLOS ONE. After careful consideration, we feel that it has merit but does not fully meet PLOS ONE’s publication criteria as it currently stands. Therefore, we invite you to submit a revised version of the manuscript that addresses the points raised during the review process.

We look forward to receiving your revised manuscript.

Kind regards,

Bing Xue, Ph.D.

Academic Editor

PLOS ONE

Journal Requirements:

"No funding"

5. We note that Figure 1 in your submission contain copyrighted images. All PLOS content is published under the Creative Commons Attribution License (CC BY 4.0), which means that the manuscript, images, and Supporting Information files will be freely available online, and any third party is permitted to access, download, copy, distribute, and use these materials in any way, even commercially, with proper attribution. For more information, see our copyright guidelines: http://journals.plos.org/plosone/s/licenses-and-copyright.

5.1.    You may seek permission from the original copyright holder of Figure 1 to publish the content specifically under the CC BY 4.0 license.

5.2.    If you are unable to obtain permission from the original copyright holder to publish these figures under the CC BY 4.0 license or if the copyright holder’s requirements are incompatible with the CC BY 4.0 license, please either i) remove the figure or ii) supply a replacement figure that complies with the CC BY 4.0 license. Please check copyright information on all replacement figures and update the figure caption with source information. If applicable, please specify in the figure caption text when a figure is similar but not identical to the original image and is therefore for illustrative purposes only.

6. We note that Figure 2 in your submission contain map images which may be copyrighted. All PLOS content is published under the Creative Commons Attribution License (CC BY 4.0), which means that the manuscript, images, and Supporting Information files will be freely available online, and any third party is permitted to access, download, copy, distribute, and use these materials in any way, even commercially, with proper attribution. For these reasons, we cannot publish previously copyrighted maps or satellite images created using proprietary data, such as Google software (Google Maps, Street View, and Earth). For more information, see our copyright guidelines: http://journals.plos.org/plosone/s/licenses-and-copyright.

6.1.    You may seek permission from the original copyright holder of Figure 2 to publish the content specifically under the CC BY 4.0 license.

6.2.    If you are unable to obtain permission from the original copyright holder to publish these figures under the CC BY 4.0 license or if the copyright holder’s requirements are incompatible with the CC BY 4.0 license, please either i) remove the figure or ii) supply a replacement figure that complies with the CC BY 4.0 license. Please check copyright information on all replacement figures and update the figure caption with source information. If applicable, please specify in the figure caption text when a figure is similar but not identical to the original image and is therefore for illustrative purposes only.

Reviewers' comments:

Reviewer's Responses to Questions

**Comments to the Author**

1. Is the manuscript technically sound, and do the data support the conclusions?

Reviewer #1: Yes

Reviewer #2: Partly

2. Has the statistical analysis been performed appropriately and rigorously? 

Reviewer #1: Yes

Reviewer #2: N/A

3. Have the authors made all data underlying the findings in their manuscript fully available?

Reviewer #1: Yes

Reviewer #2: Yes

4. Is the manuscript presented in an intelligible fashion and written in standard English?

Reviewer #1: No

Reviewer #2: No

5. Review Comments to the Author

Reviewer #1: This is an original research article, which highlights the economic valuation of natural promenades in Iran using zonal travel costs method (case study area: Gahar lake in Lorestan province). The topic is one of great interest because, at the global level, the natural promenades are an activity that increases relaxation. As human populations have grown and diversified, the demands for natural promenades have changed and increased. However, natural and financial re­sources are limited for natural promenades. Therefore, it is required to estimate the economic benefit of recreational sites for optimum allocation of scarce resources.

In the present research, 380 questionnaires were distributed among tourists via simple random sampling in tourist seasons, taking into consideration several social-economic variables of the visitors. Based on calculations, the results showed high value and importance of the zone.

The article can be accepted for publication with minor revision:

1. It would be necessary to mention the year in which this study was conducted.

2. The authors are not consistent, in some parts of the article it is used as currency Toomans (eg line 23 and 24; line 253, line 275), then Rials (eg line 200 and 201). This is confusing, it is recommended to use a single type of currency, preferably of international circulation, for example, euro or dollar.

3. The authors should use a separator (comma or point) for thousands, tens of thousands, hundreds of thousands or millions (currencies, people, hectares) (eg line 24, 71, 200, 201, line 179,180, 181, and line 103).

4. It would be appropriate to mention in the article what method it was used to determine the sample size (380 questionnaires) (eg The Cochran function).

5. The same results are presented in Table 1 and Figure 3, the authors are advised to choose a single form to present these results.

6. In Table 2 and lines 217, 218, and 219, the authors are not consistent (eg Monthly earnings= income; monthly income level). It is recommended to use a single type of description of variables.

7. It is recommended to use the dot instead of the backslash in table 2 (as Equation 2 continues with a dot).

8. It is recommended to complete the sentence from line 279.

9. The conclusions are too general. Rewriting the conclusions part, by summarizing the obtained results.

10. It is recommended to improve English.

Reviewer #2: General comment:

This is case study of the analysis focusing Economic Valuation of Natural Promenades in Iran. The novelty of this research and research framework is unclear. To be publishable, the paper will need to be massive rewritten about following points.

216-219

This paragraph should be moved to the methods section. And this part was repeats the previous sentence 169-173. The author should rearrange and simplify.

158-159

Please indicate which 6 different education levels and which 7 income ranges were considered.

Table 1 and Figure 3 show the same thing. There is no need to keep both, one of them is enough.

253, 261-262, 266-269

For comparison purposes, please keep the same unit, the unit dollar used in introduction part.

283-284

Most visitors of Gahar lake come to the zone firstly as groups and secondly as families which can be a proof for the high security of this region. Why the larger number means much more safe? Other’s research? From your questionnaire items cannot give such conclusion.

The Results part was too simple, just list the result, just listed the results, lack of cause analysis, why such results appear? What is the point of such results.

Equation (2): VR=(-9.88-1.41TC+0.000751S+2.62I-249 3.12A+2.49E)

Why space is positive？？

in 213-214 white “the average trip cost also goes higher and the largest number of people who visit the site have less expense”. And in 285-286 white “There is a reverse relationship between the distance passed and the number of travels, so as the distance goes higher, the number of references decreases.” It is a deadly contradictions to the Equation (2) ！！

Language and presentation: The manuscript is extremely confusing to read both because of the quality of language but also because of how experimental setup and results are presented. Some sentence appears again and again.

6. PLOS authors have the option to publish the peer review history of their article (what does this mean?). If published, this will include your full peer review and any attached files.

Reviewer #1: No

Reviewer #2: No

---

## [Author Response · Author response to Decision Letter 0]

29 Sep 2020

Dear Respectful Editor,

Thank you very much for the comments. We’ve done our best to provide appropriate answers to your concerns. Please find the following answers to each of the issues raised:

our submission PONE-D-20-11207R1 - [EMID:5bce6cacc1521045] ============================================= IMPORTANT: PLEASE DO NOT REPLY TO THIS EMAIL If you are unable to complete any points that are requested in this email, please explain why in the "Enter Comments" tab of the online submission form prior to re-submitting your manuscript. This will enable us to promptly assess your response and progress your manuscript to an Academic Editor at the earliest opportunity. ============================================= PONE-D-20-11207R1 Economic Valuation of Natural Promenades in Iran Using Zonal Travel Costs Method (Case study area: Gahar Lake in Lorestan Province in Western Iran) Dr Maryam Morovati Dear Dr Morovati, Thank you for submitting your manuscript entitled "Economic Valuation of Natural Promenades in Iran Using Zonal Travel Costs Method (Case study area: Gahar Lake in Lorestan Province in Western Iran)" to PLOS ONE. Your manuscript files have been checked in-house but before we can proceed we need you to address the following issues: 1) Please provide additional details regarding participant consent. In the ethics statement in the Methods and online submission information, please ensure that you have specified (1) whether consent was informed and (2) what type you obtained (for instance, written or verbal, and if verbal, how it was documented and witnessed). If your study included minors, state whether you obtained consent from parents or guardians. If the need for consent was waived by the ethics committee, please include this information. 

[Answer]: Some information on the consent of the participants was added to the text-body.

Before distributing the questionnaires, the research objectives were explained orally to the participants. They were rest-assured that participating in the survey was completely voluntary and that they could easily resign without any reprimand if they did not wish to continue cooperating after studying the questions or at any stage. The questionnaires were anonymous and participants were assured that the information would keep strictly confidential. The work was carried out after obtaining the necessary permission from the Department of Environment under the supervision of the park manager.

2) Thank you for stating the following financial disclosure: "No funding" At this time, please address the following queries: Please clarify the sources of funding (financial or material support) for your study. List the grants or organizations that supported your study, including funding received from your institution. State what role the funders took in the study. If the funders had no role in your study, please state: “The funders had no role in study design, data collection and analysis, decision to publish, or preparation of the manuscript.” If any authors received a salary from any of your funders, please state which authors and which funders. If you did not receive any funding for this study, please state: “The authors received no specific funding for this work.” Please include your amended statements within your cover letter; we will change the online submission form on your behalf.

[Answer]: since the study is a self-funded research with no financial support from funding agencies in public, commercial, or not-for-profit sectors. Therefore, the sentence “The authors received no specific funding for this work” was added to the cover letter. 

 3) We note that Figure 1 in your submission contain copyrighted images. All PLOS content is published under the Creative Commons Attribution License (CC BY 4.0), which means that the manuscript, images, and Supporting Information files will be freely available online, and any third party is permitted to access, download, copy, distribute, and use these materials in any way, even commercially, with proper attribution. For more information, see our copyright guidelines: http://journals.plos.org/plosone/s/licenses-and-copyright. We require you to either (1) present written permission from the copyright holder to publish these figures specifically under the CC BY 4.0 license, or (2) remove the figures from your submission: 1. You may seek permission from the original copyright holder of Figure 1 to publish the content specifically under the CC BY 4.0 license. We recommend that you contact the original copyright holder with the Content Permission Form (http://journals.plos.org/plosone/s/file?id=7c09/content-permission-form.pdf) and the following text: “I request permission for the open-access journal PLOS ONE to publish XXX under the Creative Commons Attribution License (CCAL) CC BY 4.0 (http://creativecommons.org/licenses/by/4.0/). Please be aware that this license allows unrestricted use and distribution, even commercially, by third parties. Please reply and provide explicit written permission to publish XXX under a CC BY license and complete the attached form.” Please upload the completed Content Permission Form or other proof of granted permissions as an "Other" file with your submission. In the figure caption of the copyrighted figure, please include the following text: “Reprinted from [ref] under a CC BY license, with permission from [name of publisher], original copyright [original copyright year].” 2. If you are unable to obtain permission from the original copyright holder to publish these figures under the CC BY 4.0 license or if the copyright holder’s requirements are incompatible with the CC BY 4.0 license, please either i) remove the figure or ii) supply a replacement figure that complies with the CC BY 4.0 license. Please check copyright information on all replacement figures and update the figure caption with source information. If applicable, please specify in the figure caption text when a figure is similar but not identical to the original image and is therefore for illustrative purposes only. 

[Answer]: we hereby declare that Figure 1 has not been copied from any websites, database, and the like. We, authors, has prepared the figure. 

4) We note that Figure 2 in your submission contain map images which may be copyrighted. All PLOS content is published under the Creative Commons Attribution License (CC BY 4.0), which means that the manuscript, images, and Supporting Information files will be freely available online, and any third party is permitted to access, download, copy, distribute, and use these materials in any way, even commercially, with proper attribution. For these reasons, we cannot publish previously copyrighted maps or satellite images created using proprietary data, such as Google software (Google Maps, Street View, and Earth). For more information, see our copyright guidelines: http://journals.plos.org/plosone/s/licenses-and-copyright. We require you to either (1) present written permission from the copyright holder to publish these figures specifically under the CC BY 4.0 license, or (2) remove the figures from your submission: 1. You may seek permission from the original copyright holder of Figure 2 to publish the content specifically under the CC BY 4.0 license. We recommend that you contact the original copyright holder with the Content Permission Form (http://journals.plos.org/plosone/s/file?id=7c09/content-permission-form.pdf) and the following text: “I request permission for the open-access journal PLOS ONE to publish XXX under the Creative Commons Attribution License (CCAL) CC BY 4.0 (http://creativecommons.org/licenses/by/4.0/). Please be aware that this license allows unrestricted use and distribution, even commercially, by third parties. Please reply and provide explicit written permission to publish XXX under a CC BY license and complete the attached form.” Please upload the completed Content Permission Form or other proof of granted permissions as an "Other" file with your submission. In the figure caption of the copyrighted figure, please include the following text: “Reprinted from [ref] under a CC BY license, with permission from [name of publisher], original copyright [original copyright year].” 2. If you are unable to obtain permission from the original copyright holder to publish these figures under the CC BY 4.0 license or if the copyright holder’s requirements are incompatible with the CC BY 4.0 license, please either i) remove the figure or ii) supply a replacement figure that complies with the CC BY 4.0 license. Please check copyright information on all replacement figures and update the figure caption with source information. If applicable, please specify in the figure caption text when a figure is similar but not identical to the original image and is therefore for illustrative purposes only. The following resources for replacing copyrighted map figures may be helpful: USGS National Map Viewer (public domain): http://viewer.nationalmap.gov/viewer/ The Gateway to Astronaut Photography of Earth (public domain): http://eol.jsc.nasa.gov/sseop/clickmap/ Maps at the CIA (public domain): https://www.cia.gov/library/publications/the-world-factbook/index.html and https://www.cia.gov/library/publications/cia-maps-publications/index.html NASA Earth Observatory (public domain): http://earthobservatory.nasa.gov/ Landsat: http://landsat.visibleearth.nasa.gov/ USGS EROS (Earth Resources Observatory and Science (EROS) Center) (public domain): http://eros.usgs.gov/# Natural Earth (public domain): http://www.naturalearthdata.com/

[Answer]: we should declare that Figure 1 has not been copied from any websites, database, and the like. We, authors, has prepared the figure. 

5) Thank you for updating your data availability statement. You note that your data are available within the Supporting Information files, but no such files have been included with your submission. At this time we ask that you please upload your minimal data set as a Supporting Information file, or to a public repository such as Figshare or Dryad. Please also ensure that when you upload your file you include separate captions for your supplementary files at the end of your manuscript. 

[Answer]: we changed our choice of the data availability statement. Since we cannot provide you with the data of the questionnaires according to our ethical commitment; therefore, we express our disagreement with the data availability.

Dear Editor,

We are thankful for the valuable detailed comments of the reviewers on the manuscript. The suggestions are quite helpful for us and we incorporate them in the revised paper. As below, on behalf of my co-authors, I would like to clarify the points raised by the Reviewers and we hope the Reviewers and the Editors will be satisfied with our responses to the ‘comments’ and the revisions for the original manuscript. 

Thanks and Best Regards!

Morovati

Reviewers' comments:

Reviewer #1: This is an original research article, which highlights the economic valuation of natural promenades in Iran using zonal travel costs method (case study area: Gahar lake in Lorestan province). The topic is one of great interest because, at the global level, the natural promenades are an activity that increases relaxation. As human populations have grown and diversified, the demands for natural promenades have changed and increased. However, natural and financial re¬sources are limited for natural promenades. Therefore, it is required to estimate the economic benefit of recreational sites for optimum allocation of scarce resources.

In the present research, 380 questionnaires were distributed among tourists via simple random sampling in tourist seasons, taking into consideration several social-economic variables of the visitors. Based on calculations, the results showed high value and importance of the zone.

The article can be accepted for publication with minor revision:

1. It would be necessary to mention the year in which this study was conducted.

[Answer]: thank you for the comment. The year of the study was 2018 as added into the text-body, under the subtitle of “Research Methodology” 

2. The authors are not consistent, in some parts of the article it is used as currency Toomans (eg line 23 and 24; line 253, line 275), then Rials (eg line 200 and 201). This is confusing, it is recommended to use a single type of currency, preferably of international circulation, for example, euro or dollar.

[Answer]: thank you for the comment. The format was standardized as in USD. 

3. The authors should use a separator (comma or point) for thousands, tens of thousands, hundreds of thousands or millions (currencies, people, hectares) (eg line 24, 71, 200, 201, line 179,180, 181, and line 103).

[Answer]: thank you for the comment. The presentation of the values was modified as recommended. 

4. It would be appropriate to mention in the article what method it was used to determine the sample size (380 questionnaires) (eg The Cochran function).

[Answer]: thank you for the comment. The sample size was determined based on Cochran function. 

5. The same results are presented in Table 1 and Figure 3, the authors are advised to choose a single form to present these results.

[Answer]: thank you for the comment. Table 1 was deleted. 

For comparison purposes, please keep the same unit, the unit dollar used in introduction part.

283-284

[Answer]: thank you for the comment. The format for presenting the values was standardized.

Most visitors of Gahar lake come to the zone firstly as groups and secondly as families which can be a proof for the high security of this region. Why the larger number means much more safe? Other’s research? From your questionnaire items cannot give such conclusion.

[Answer]: thank you for the comment. The science was modified as follow:

“The high number of visitors can be due to reasons such as the unique beauty of the landscape and its high security. Security is the most important and main criterion for tourists all over the world. No matter how unique the tourist areas are, the necessary condition for attracting tourists is first and foremost the existence of security in the tourist centers.” 

The Results part was too simple, just list the result, just listed the results, lack of cause analysis, why such results appear? What is the point of such results.

[Answer]: thank you for the comment. the results and discussion part have been modified extensively. The results have been supported by concrete reasoning and compared with the following updated relevant studies:

“Ofori R.O. , Rouleau M.D. (2020). Willingness to pay for invasive seaweed management: Understanding how high and low income households differ in Ghana. Ocean & Coastal Management, Volume 192, 1 July 2020, 105224

Xu F., Wang Y., Xiang N., Tian J., Chen L. (2020). Uncovering the willingness-to-pay for urban green space conservation: A survey of the capital area in China. Resources, Conservation and Recycling, Volume 162, November 2020, 105053.

Tavárez H., Elbakidze L. (2019). Valuing recreational enhancements in the San Patricio Urban Forest of Puerto Rico: A choice experiment approach. Forest Policy and Economics, Volume 109, December 2019, 102004

Lankia T., Neuvonen M., Pouta E., Sievänen T. (2014).Willingness to contribute to the management of recreational quality on private lands in Finland. Journal of Forest Economics, Volume 20, Issue 2, April 2014, Pages 141-160

Cavasos K.E., Bhat M.G. (2020).Impact of risk perception on household willingness-to-pay to restock the threatened staghorn coral. Ocean & Coastal Management, Volume 193, 1 August 2020, 105244

Tian Y., Wu H., Zhang G., Wang L., Zheng D., Li S. (2020).Perceptions of ecosystem services, disservices and willingness-to-pay for urban green space conservation. Journal of Environmental Management, Volume 260, 15 April 2020, 110140

Enriquez-Acevedo T., Botero C.M., Cantero-Rodelo R., Pertuz A., Suarez A. (2018). Willingness to pay for Beach Ecosystem Services: The case study of three Colombian beaches. Ocean & Coastal Management, Volume 161, 1 July 2018, Pages 96-104” 

Equation (2): VR=(-9.88-1.41TC+0.000751S+2.62I-249 3.12A+2.49E)

Why space is positive？？

in 213-214 white “the average trip cost also goes higher and the largest number of people who visit the site have less expense”. And in 285-286 white “There is a reverse relationship between the distance passed and the number of travels, so as the distance goes higher, the number of references decreases.” It is a deadly contradictions to the Equation (2) ！！

[Answer]: thank you for the comment. it seems to be a typo mistake. The equation was corrected. 

Language and presentation: The manuscript is extremely confusing to read both because of the quality of language but also because of how experimental setup and results are presented. Some sentence appears again and again.

 [Answer]: thank you for the comment. The English of the paper was revised by a professional English language editor to fix grammatical errors. The duplicate content has been removed.

---

## [Decision Letter · Decision Letter 1]

14 Oct 2020

Economic Valuation of Natural Promenades in Iran Using Zonal Travel Costs Method (Case study area: Gahar Lake in Lorestan Province in Western Iran)

PONE-D-20-11207R1

Dear Dr. Morovati,

We’re pleased to inform you that your manuscript has been judged scientifically suitable for publication and will be formally accepted for publication once it meets all outstanding technical requirements.

Kind regards,

Bing Xue, Ph.D.

Academic Editor

PLOS ONE

Additional Editor Comments (optional):

Reviewers' comments:

Reviewer's Responses to Questions

**Comments to the Author**

1. If the authors have adequately addressed your comments raised in a previous round of review and you feel that this manuscript is now acceptable for publication, you may indicate that here to bypass the “Comments to the Author” section, enter your conflict of interest statement in the “Confidential to Editor” section, and submit your "Accept" recommendation.

Reviewer #1: All comments have been addressed

2. Is the manuscript technically sound, and do the data support the conclusions?

Reviewer #1: Yes

3. Has the statistical analysis been performed appropriately and rigorously? 

Reviewer #1: Yes

4. Have the authors made all data underlying the findings in their manuscript fully available?

Reviewer #1: Yes

5. Is the manuscript presented in an intelligible fashion and written in standard English?

Reviewer #1: Yes

6. Review Comments to the Author

Reviewer #1: The authors have taken into account all the recommendations, and the article is significantly improved.

7. PLOS authors have the option to publish the peer review history of their article (what does this mean?). If published, this will include your full peer review and any attached files.

Reviewer #1: No

---

## [Editor Report · Acceptance letter]

26 Oct 2020

PONE-D-20-11207R1 

Economic Valuation of Natural Promenades in Iran Using Zonal Travel Costs Method (Case study area: Gahar Lake in Lorestan Province in Western Iran) 

Dear Dr. Morovati:

I'm pleased to inform you that your manuscript has been deemed suitable for publication in PLOS ONE. Congratulations! Your manuscript is now with our production department. 

Kind regards, 

on behalf of

Professor Bing Xue 

Academic Editor

PLOS ONE